# Prediction of Primary Tumor Sites in Spinal Metastases Using a ResNet-50 Convolutional Neural Network Based on MRI

**DOI:** 10.3390/cancers15112974

**Published:** 2023-05-30

**Authors:** Ke Liu, Siyuan Qin, Jinlai Ning, Peijin Xin, Qizheng Wang, Yongye Chen, Weili Zhao, Enlong Zhang, Ning Lang

**Affiliations:** 1Department of Radiology, Peking University Third Hospital, Beijing 100191, China; 2011210430@bjmu.edu.cn (K.L.); 1510301315@pku.edu.cn (S.Q.); xinpeijin1997@bjmu.edu.cn (P.X.); wangqizheng96@163.com (Q.W.); chenyongye1995@163.com (Y.C.); m17853252989@163.com (W.Z.); 18811728521@163.com (E.Z.); 2Department of Informatics, King’s College London, London WC2B 4BG, UK; jinlai7@foxmail.com

**Keywords:** spinal metastases, convolutional neural network, MRI

## Abstract

**Simple Summary:**

Spinal metastases are a common occurrence, and many patients do not have a clear history of primary tumors when diagnosed with spinal metastases. Patients with cancer of unknown primary often undergo comprehensive and invasive diagnostic work-ups, which can be expensive and time-consuming. To narrow down the search for primary tumor sites in spinal metastases patients with cancer of unknown primary, an artificial intelligence model that can assess tumor origin on MRI may be beneficial. Our work builds upon the considerable interest in investigating the feasibility of using a ResNet-50 convolutional neural network model based on MRI in predicting primary tumor sites in spinal metastases. In this preliminary study, for the 5-class classification of spinal metastases originating from the lung, kidney, prostate, and mammary or thyroid glands, the AUC-ROC and top-1 accuracy of the ResNet-50 model were 0.77 and 52.97%, respectively. Therefore, we believe that the ResNet-50 CNN model for predicting primary tumor sites in spinal metastases using MRI has the potential to help prioritize examinations and treatments in cases of unknown primary for radiologists and oncologists.

**Abstract:**

We aim to investigate the feasibility and evaluate the performance of a ResNet-50 convolutional neural network (CNN) based on magnetic resonance imaging (MRI) in predicting primary tumor sites in spinal metastases. Conventional sequences (T1-weighted, T2-weighted, and fat-suppressed T2-weighted sequences) MRIs of spinal metastases patients confirmed by pathology from August 2006 to August 2019 were retrospectively analyzed. Patients were partitioned into non-overlapping sets of 90% for training and 10% for testing. A deep learning model using ResNet-50 CNN was trained to classify primary tumor sites. Top-1 accuracy, precision, sensitivity, area under the curve for the receiver-operating characteristic (AUC-ROC), and F1 score were considered as the evaluation metrics. A total of 295 spinal metastases patients (mean age ± standard deviation, 59.9 years ± 10.9; 154 men) were evaluated. Included metastases originated from lung cancer (*n* = 142), kidney cancer (*n* = 50), mammary cancer (*n* = 41), thyroid cancer (*n* = 34), and prostate cancer (*n* = 28). For 5-class classification, AUC-ROC and top-1 accuracy were 0.77 and 52.97%, respectively. Additionally, AUC-ROC for different sequence subsets ranged between 0.70 (for T2-weighted) and 0.74 (for fat-suppressed T2-weighted). Our developed ResNet-50 CNN model for predicting primary tumor sites in spinal metastases at MRI has the potential to help prioritize the examinations and treatments in case of unknown primary for radiologists and oncologists.

## 1. Introduction

Due to the red bone marrow serving an optimal metastatic niche [1], the spine is the third most common site for distant metastases of malignant tumors, second only to the lungs and liver [2]. More than 50% of patients with malignant tumors will develop spinal metastases [3]. The proportions of some common primary sites for spinal metastases have been reported as follows: 13.9–37.0% for lung cancer, 16.2–16.5% for unknown origin, 6.4–16.9% for breast cancer, 4.4–14.8% for prostate cancer %, 6.1–12.0% for kidney cancer, and 2.5–3.1% for thyroid cancer [4,5,6,7]. A study showed that 73.3% of patients had no clear history of primary tumors when diagnosed with spinal metastases. Most saw a doctor because of local pain or compression fractures [6]. However, the primary tumor site is one of the most important factors affecting survival in some of the published prognostic scoring systems (Tokuhashi, Tomita, van der Linden, Bauer) for spinal metastases [8,9]. Therefore, differential diagnosis of the primary tumor site of spinal metastases is crucial for patient prognosis prediction and treatment decision-making [10].

Patients with cancer of unknown primary often undergo comprehensive diagnostic work-ups including imaging, histopathology, molecular diagnosis, serum tumor markers, and endoscopy to determine the occult primary site [11,12]. However, these work-ups can be expensive, time-consuming, and invasive. Furthermore, in 8.1–16.5% of patients with spinal metastases, these work-ups may fail to detect a primary tumor [4,5,6]. MRI is the suggested primary imaging modality for detecting and diagnosing spinal metastases because it can detect early bone marrow deposits and provide essential information on the characterization of the levels of involvement [13,14].

Deep learning algorithms, particularly the convolutional neural network (CNN), have rapidly become a methodology of choice for image classification in medical images [15]. Recently, studies have shown that deep learning models using ResNet-50 CNN based on MRI not only provide a feasible method for the differential diagnosis of benign and malignant vertebral fracture [16] but also perform well in multi-classification tasks of tumors [17]. In addition, some studies have shown that deep learning models can predict the mutational status of tumor genes on MRI [18,19]. To narrow the search for primary tumor sites in spinal metastases patients with cancer of unknown primary, an artificial intelligence model that could assess tumor origin on MRI may be beneficial.

This study aimed to investigate the feasibility and evaluate the performance of a ResNet-50 CNN model based on MRI in predicting primary tumor sites in spinal metastases.

## 2. Materials and Methods

This retrospective study was approved by and registered with Peking University Third Hospital Medical Science Research Ethics Committee, and the requirement of informed consent was waived.

### 2.1. Study Population

MRIs of patients with clinical suspicion of spinal metastases at Peking University Third Hospital from August 2006 to August 2019 were retrospectively analyzed. The inclusion criteria were as follows: (1) patients with pathologically confirmed spinal metastases; and (2) sagittal T1, T2, and fat-suppressed T2-weighted sequences of the spine were in full. The exclusion criteria were (1) the primary tumor sites was not confirmed by a pathology report; (2) surgery was performed for the lesion of the spinal metastases before MRI and CT examinations; and (3) the number of patients at this primary tumor site is less than 25 (Figure 1).

### 2.2. Imaging Acquisition

Sagittal T1-weighted, T2-weighted, and fat-suppressed T2-weighted MRI sequences of all patients were acquired with a 1.5-T MRI unit (Magnetom Sonata, Siemens Healthcare, Erlangen, Germany; Optima MR360, GE Medical Systems, LLC, Chicago, IL, USA); 3.0-T MRI unit (Magnetom Trio, Siemens Healthcare, Erlangen, Germany; or Signa HDx, GE Medical Systems, LLC; or Discovery MR750, GE Medical Systems, LLC; or Discovery MR750w, GE Healthcare Japan Corporation, Tokyo, Japan). Imaging protocols and scanning parameters of cervical, thoracic, lumbar, and sacral vertebras are shown in Appendix A. Sequence parameters varied among the different MR units, reflecting the heterogeneity of image data in clinical practice.

Digital imaging and communications in medicine images of all identified patients were exported from the picture archiving and communication system. Subsequently, protected health information was entirely removed from all images. Finally, the images were converted from digital imaging and communications in medicine format to JPG format. 

### 2.3. Data Pre-Processing

The input size is 512 × 512 and the channel number is 3 for the RGB color image. Before feeding into the networks, the pixel values are rescaled to a floating number between 0 to 1. To avoid the variant of different samples, we perform sample-wise standard normalization. To extend the size of the dataset, we apply image augmentation techniques including horizontal and vertical shifting, horizontal flipping, and greyscale inverting.

One of the main problems of the dataset is category unbalance. The number of “lung cancer” samples is several times more than samples of other labels. Without invention, it would cause the model to tend to predict “lung cancer” during training, which is not what we expect. To balance the samples in training, we use a statistic-based factor to give a weight to each sample based on the category it belongs to, as shown in Equation (1). The weighted binary cross entropy (WBCE) calculates the weighted sum of loss, in favor of samples from minority category. Weight $\beta_{c}$ is simply calculated using the reciprocal for the proportion of the certain category out of the whole dataset. WBCE aims to force the samples fewer in number to contribute more to the loss function, which could ensure they are valued during the training.
(1)$$WBCE\left(\hat{p},p\right)=-\left(\beta_{c}\cdot p\cdot log\left(p\right)+\left(1-\hat{p}\right)\cdot log\left(1-p\right)\right)\;where\;\beta_{c}=\frac{N}{n_{c}}$$
where $\hat{p}$ and p indicate predicted value and ground truth; $\beta_{c}$ indicates the weight used for samples within category c; N and $n_{c}$ are the number of samples in training set and in category
c
respectively.

### 2.4. Training Set and Testing Set Split

We split 90% of the images as the training set and the remaining 10% as the testing set. Here, the basic unit for splitting is the patient rather than the single image, which ensures images of one patient could not be in both the training set and testing set. The numbers of images in the training set and testing set are shown in Appendix A. 

### 2.5. Model Architecture

We propose a simple but effective architecture. Our network has three parts: backbone, feature fusion module, and feature utilization module, as shown in Figure 2.

The backbone of our model is ResNet-50. ResNet [20] is a series of convolution networks that have obtained impressive performance in image classification. ResNet-50, a 50-layer ResNet, consists of a convolutional layer, a max-pooling layer, and 16 residual blocks. Each residual block consists of one 3 × 3 convolutional layer, two 1 × 1 convolutional layers, and skips connections from inputs to outputs. The overall stride of ResNet-50 is 32, which means the width and height of features before feeding the final fully connected layer are 32 times smaller than the width and height of original images. The detailed architecture is shown in Appendix A.

The feature fusion module aims to generate features combing spatial information with rich semantic information. In this section, we apply a modified feature pyramid network [21] to fuse features of different scales. It builds a top-down feature fusion pathway separate from the original bottom-up feature extraction pathway. As specified, the fusion is conducted via an element-wise sum between the 2-times up-sampling deeper features and shallow features that aligns the number of channels. The feature fusion module repeats this process four times hierarchically to get a 4-layer pyramid of features with different scales.

The feature utilization module aims to utilize features to make the final prediction. Firstly, we conduct global pooling on each output of the feature fusion module and concatenate the flattened vectors together. Then, the concatenated features are fed into a fully connected layer with 5 output units to make the final prediction. By using softmax [22] as an activation function, the outputs are formatted as probabilities, as shown in Equation (2).
(2)$$softmax\left(s_{i}\right)=\frac{e^{x_{i}}}{{\sum^{\;}}_{i=1}^{n}e^{x_{i}}}$$

### 2.6. Training and Evaluation

The loss function is categorical cross-entropy, which is a commonly used loss function. The goal of training is to find an optimal input to obtain a smaller value of loss function than most. For the optimizer, we have used stochastic gradient descent [23] and Adam [24] in different experiments. Both optimizers are gradient-based iterative methods that find the optimal minimum of loss function depending on the value of the gradient of outputs to inputs step by step. The batch size in different experiments varies from 13 to 32. Random shuffle is also applied to search in a wider space. We add an L2 regularization on the activation function with a rate of 0.07. The interpolation method used is bilinear interpolation. The dropout rate is 0.005. 

In addition to the initial training and testing approach, we also implemented a cross-validation method to further validate the robustness of our model. Specifically, we employed 5-fold cross-validation, a widely used technique in machine learning to assess the generalizability of a model. In 5-fold cross-validation, the complete dataset was randomly partitioned into 5 equally sized subsets. Of these, four subsets were used for training the model, and the remaining subset was used for testing. This process was repeated 5 times, with each of the 5 subsets used exactly once as the testing data. The 5 results from these folds then were averaged to produce a single estimation.

We consider top-1 accuracy, precision, sensitivity, area under the curve for the receiver-operating characteristic (AUC-ROC), and F1 score as the evaluation metrics.

### 2.7. Inference and Visualization

During the inference, we apply single image inference. The category with maximum probability is determined as the predicted category. For the gender bias problem, we conduct a simple gender filtering method to ensure the outputs are not clinically meaningless. To specify, we set the probability of the category that would not happen or rarely happen in the current gender to zero and rescale the probability to a 1-sum distribution.

For the real scene, we visualize the results in a human-friendly interface. It would show the probability with some bars floating on the testing image. 

### 2.8. Prior Information from Gender and Age

Gender and age are considered important prior information for diagnosis. Therefore, apart from the CNNs, we also use machine learning models to make use of patients’ gender and age information. Our work tests with naive Bayes, support vector machine, logistic regression, k-nearest neighbors and random forests according to gender and age. The detailed methods were described in the Appendix A. The independent classifier could work as an auxiliary model for the final diagnosis. 

## 3. Results

### 3.1. Patient Characteristics

A total of 295 spinal metastases patients (154 men and 141 women), with at least 13 MRI images for each patient, originating from lung cancer (*n* = 142), kidney cancer (*n* = 50), mammary cancer (*n* = 41), thyroid cancer (*n* = 34), and prostate cancer (*n* = 28) were included. The mean age ± standard deviation at enrollment was 59.9 years ± 10.9, with a mean of 61.8 years ± 10.1 for men and 57.8 years ± 11.3 for women. In addition, the specific demographics and clinical characteristics are shown in Table 1.

### 3.2. Performance of the ResNet-50 Model

By changing the final fully connected layer, we divide the models into 5-class classifiers, 4-class classifiers, and 3-class classifiers. To be clear, the 3 (or 4)-class results are the average result among all possible combinations of 3 (or 4) classes out of 5. We use our model to perform a series of 3 (or-4)-class classification by only replacing the topmost layer. All the metrics are shown in Table 2. The results prove the effectiveness of our model. For 5-class classification, the main focus of our work, we obtain 52.97% top-1 accuracy and 0.77 AUC-ROC. The absolute value is not high because of our problem setting. It is significant progress to make predictions automatically, given that even experienced radiologists cannot make trusted diagnostics. Additionally, while narrowing the problem to smaller categories, the model shows better performance. For 3-class classification, it can obtain 67.16% top-1 accuracy and 0.85 AUC-ROC. 

Studies have shown that different MRI sequences play different roles in assessing bone metastases [25,26]. Therefore, the AUC-ROC of the subsets of T1-weighted, T2-weighted, and fat-suppressed T2-weighted MRI sequences, respectively, reached 0.74, 0.75, and 0.70. The other evaluation metrics are shown in Table 2. In addition, it is possible that we might want to judge the category with images from only one sequence. Our model can maintain a competitive AUC-ROC value no matter which subset is used. It shows the generalization power of our model to some extent.

The performance of our ResNet-50 convolutional neural network model was evaluated using a 5-fold cross-validation procedure. Detailed performance metrics for each fold are provided in Table 3. The evaluation results are stable among all folds. On average, across all five folds, the model demonstrated a degree of consistency in its ability to predict primary tumor sites in spinal metastases.

Gender and age are useful, commonly used hand-crafted features for medical diagnosis based on previous research. However, in this paper, we do not integrate gender and age as features with our model. This is because redundant features may cause damage to the overall result based on our experiments, as shown in Table 4. Here, we are not proving these features are meaningless but want to show that our model can perform very well without these features. Convolution neural networks make predictions based on high-dimensional information that may contain features known and unknown to humans. The gender and age could be redundant here because the model may already have sufficient information to make the correct prediction.

### 3.3. Model Interpretation and Examples

Some visualization examples are shown in Figure 3 and Figure 4. The green label is the real category that would be hidden during inference. The orange label shows the highest probability, and the white label shows the probabilities of other categories. The visualization results give an intuitive point for humans. We visualize the results using a set of transparent bars floating on the examined images.

Figure 5 illustrates the loss value of the 5-class classifier among the training set and test(validation) set during training. It is clear that the loss value dropped on both the training set and test set. This indicates that, during the training, our model learns how to make correct classification gradually.

## 4. Discussion

When spinal metastatic cancer is suspected, finding and confirming the primary lesion becomes the most important task for treatment planning. However, finding the primary site is challenging, time-consuming, and requires multiple examinations. Our results suggest that a deep learning model using ResNet-50 trained on MRI data can help predict the metastatic tumor type for patients with spinal metastases. We obtained 52.97% top-1 accuracy and 0.77 AUC-ROC for 5-class classification. Additionally, our model can maintain a competitive AUC-ROC value no matter which MRI sequence subset is used. The visualization results can support the human expert in prioritizing the examinations of probable primary tumor sites. 

The feasibility of radiomics or deep learning models based on CT or MR images in predicting origins for cancers of unknown primary site has been demonstrated for brain metastases and liver metastases. Avi Ben-Cohen et al. proposed a 4-class classification automated system for the categorization of liver metastases into primary cancer sites. Additionally, the top-1 accuracy result when using CNN features is 50% [27]. With a similar problem setting, we obtained a top-1 accuracy of 58% based on the more readily available non-enhanced MRI compared to enhanced CT images. Helge C. Kniep et al.’s proposed classifier achieved AUCs between 0.64 and 0.82 in a 5-class radiomics model of brain metastases into primary cancer sites [28]. This performance is comparable to our result of 0.77 average AUC-ROC for 5-class classification. However, this radiomics model is trained on enhanced MR images. Furthermore, compared to these previous studies, we do not need human experts to delineate regions of interest (ROI) manually. 

In previous studies [27,28], the knowledge of the age and gender of the patient can add a significant improvement to the results of predicting metastatic tumor types. In this paper, we do not integrate gender and age as features with the features obtained from the CNN. This is because we find no clear increase from doing so. This suggests that gender and age information is abundant in the features that the model has learned. Possible reasons are that the limited number of samples hides the data distribution, or our model has extracted their information from images. 

Not only images with clearly observed tumor lesions but also images without obvious tumor lesions are included in the training set. This is because, although we cannot recognize lesions in these images, they do come from confirmed cases. It would be too hasty to remove significant quantities of images without evidence that they are useless, as we may waste chances to extract information that radiologists have not considered important. Previous studies [29,30] have shown that radiologists can struggle to see patterns in the surrounding tissues based on region-level features selected by visual inspection; deep learning methods can identify some connections between the morphology changes and disease properties by learning pixel-level features. Rather than import human interference, we believe it is better to make the best use of the automatic feature extraction power of CNN, especially for this open question. 

Our study had the following general and study-specific limitations: First, a major bottleneck in developing and using deep learning methods for medical image classification is the lack of large and class-balanced datasets required for training deep learning models [31,32,33]. Although we applied image augmentation techniques and used a statistic-based factor to balance the samples in training, we only considered metastases derived from the five most common primary sites of origin as an initial study. The 3-class classification results support the idea that our model could still perform well even with a more severe unbalance caused by the decreased number of categories. Second, as a single-center study, there is a lack of a proper independent external validation set. However, the varied sequence parameters among the different MRI units also reflect the heterogeneity of image data to a certain extent. Third, compared to contrast-enhanced T1-weighted sequences and functional sequences, such as Dixon, diffusion-weighted imaging (DWI), and dynamic contrast-enhanced (DCE) sequences, conventional sequences provide limited information for tissue heterogeneity and the tumor microenvironment. However, conventional sequences are included for almost all standard MRI protocols, so the developed ResNet-50 model is generalizable and feasible for application in clinical practice. More advanced sequences may be used for further prospective studies. Fourth, our model is good but not highly satisfactory at this moment, as shown by the evaluation metrics. The problem we are facing is an unsolved one for even experienced doctors, so our tool is useful for giving a reasonable prediction. It could be more precise, but we could also accept it because it can potentially save a lot of time for a doctor to arrange further tests at the highly suspected tumor site rather than making a random selection. Our CNN model is a primary proposal to solve this challenging problem, which clearly has the potential to be improved in the future. We will focus on improving the performance in our following research. Finally, diagnoses and treatment decisions are made at the patient-level, but we have not aggregated image-level classification and produced patient-level diagnoses. Integrating uncertainty information about image predictions in aggregation models will result in higher uncertainty measures for false patient classifications [34]. Therefore, we need to develop an effective aggregation model to overcome this limitation.

## 5. Conclusions

In conclusion, the ResNet-50 deep learning model we developed provides a noninvasive automatic methodology for the categorization of the metastatic tumor type for patients with spinal metastases. This model can help prioritize examinations and treatments in cases of unknown primary sites.

## Figures and Tables

**Figure 1 cancers-15-02974-f001:**
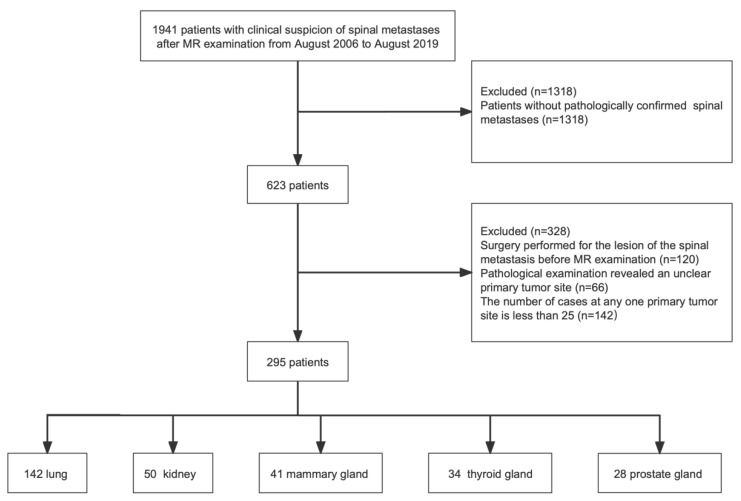
Flowchart demonstrating the inclusion and exclusion criteria for this study.

**Figure 2 cancers-15-02974-f002:**
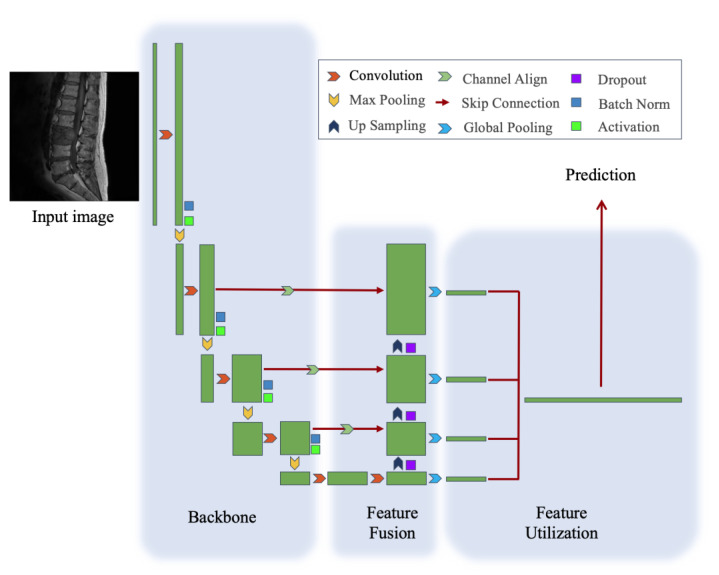
Architecture of the model, which includes backbone, feature fusion module, and feature utilization module.

**Figure 3 cancers-15-02974-f003:**
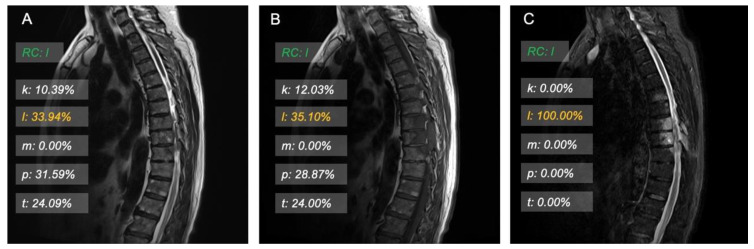
Images in a 75-year-old man who was correctly predicted as having spinal metastases that originated from lung cancer by the ResNet50 model. (**A**) Unenhanced sagittal T2-weighted image. (**B**) Unenhanced sagittal T1-weighted image. (**C**) Unenhanced sagittal fat-suppressed T2-weighted image. RC = real category, k = probability of originating from kidney cancer, l = probability of originating from lung cancer, m = probability of originating from mammary cancer, p = probability of originating from prostate cancer, t = probability of originating from thyroid cancer.

**Figure 4 cancers-15-02974-f004:**
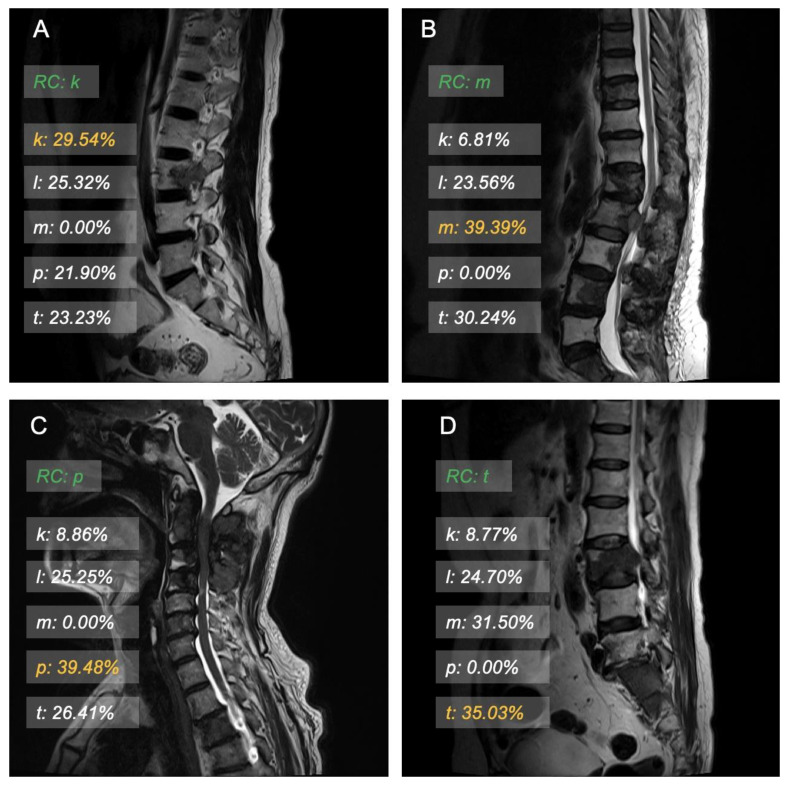
(**A**) Unenhanced sagittal T2-weighted image in a 44-year-old man who was correctly predicted as having spinal metastases originating from kidney cancer by the ResNet50 model. (**B**) Unenhanced sagittal T2-weighted image in a 61-year-old woman who was correctly predicted as having spinal metastases originating from mammary cancer by the ResNet50 model. (**C**) Unenhanced sagittal T2-weighted image in a 69-year-old man who was correctly predicted as having spinal metastases originating from prostate cancer by the ResNet50 model. (**D**) Unenhanced sagittal T2-weighted image in a 67-year-old woman who was correctly predicted as having spinal metastases originating from thyroid cancer by the ResNet50 model. RC = real category, k = probability of originating from kidney cancer, l = probability of originating from lung cancer, m = probability of originating from mammary cancer, p = probability of originating from prostate cancer, t = probability of originating from thyroid cancer.

**Figure 5 cancers-15-02974-f005:**
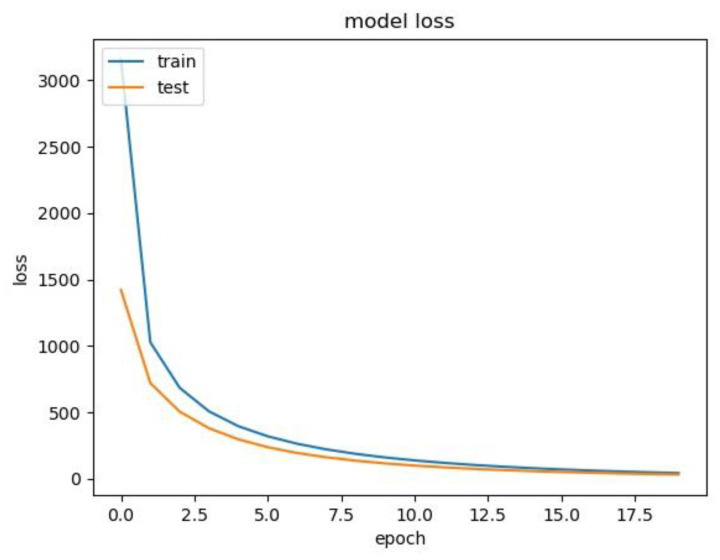
Loss curve of 5-class classifier among the training set and test (validation) set.

**Table 1 cancers-15-02974-t001:** Demographic and clinical characteristics of patients (N = 295).

	Lung Cancer (*n* = 142)	Kidney Cancer (*n* = 50)	Mammary Cancer (*n* = 41)	Thyroid Cancer (*n* = 34)	Prostate Cancer (*n* = 28)
Sex					
M	76	38	0	12	28
F	66	12	41	22	0
Mean age (y) *					
M	61.4 ± 9.7 (38–81)	58.6 ± 9.0 (37–72)	-	60.8 ± 6.8 (46–72)	68.0 ± 11.0 (39–84)
F	62.4 ± 9.5 (37–82)	56.5 ± 12.6 (30–75)	52.1 ± 8.7 (34–73)	55.3 ± 13.3 (27–76)	-
Scan location					
Cervical spine	49	24	13	20	7
Thoracic spine	25	12	10	4	11
Lumbar spine	65	14	18	10	10
Sacral spine	3	0	0	0	0

* Numbers in parentheses represent the data range.

**Table 2 cancers-15-02974-t002:** Performance results of 5-class classifiers, 4-class classifiers, and 3-class classifiers.

Model	Top-1 Accuracy (%)	Precision (%)	Sensitivity (%)	Specificity (%)	AUC-ROC	F1 Score
5-class	52.97 (52.08~53.86)	59.84 (59.18~60.50)	48.56 (47.95~49.17)	61.81 (57.23~66.39)	0.77 (0.76~0.77)	0.54 (0.53~0.54)
T1WS	49.05 (48.51~49.59)	100.00 (99.99~100)	23.09 (22.30~23.88)	65.50 (62.78~68.22)	0.74 (0.73~0.75)	0.38 (0.37~0.39)
T2WS	41.83 (41.09~42.57)	43.73 (42.54~44.92)	50.00 (49.27~50.73)	49.77 (47.28~52.06)	0.75 (0.74~0.76)	0.47 (0.46~0.47)
T2WS-FS	36.32 (35.36~37.28)	45.83 (45.28~46.38)	40.01 (39.11~40.91)	68.00 (65.87~70.13)	0.70 (0.69~0.71)	0.43 (0.42~0.44)
4-class	58.46 (57.78~59.14)	61.82 (61.33~62.31)	57.13 (56.67~57.59)	80.77 (75.69~85.85)	0.81 (0.80~0.82)	0.59 (0.59~0.59)
3-class	67.16 (66.22~68.12)	68.99 (68.04~68.12)	66.91 (66.37~67.45)	83.97 (77.42~90.52)	0.85 (0.84~0.86)	0.68 (0.67~0.69)

Note. T2WS = T2-weighted sequence. T1WS = T1-weighted sequence. T2WS-FS = fat-suppressed T2 sequence. AUC-ROC = area under the curve for the receiver-operating characteristic. The range is calculated based on the 95% confidence interval.

**Table 3 cancers-15-02974-t003:** 5-fold cross evaluation results of 5-class classifiers.

Model	Top-1 Accuracy (%)	Precision (%)	Sensitivity (%)	Specificity (%)	AUC-ROC	F1 Score
Fold 1	58.68	61.27	53.42	67.14	0.80	0.56
Fold 2	56.06	60.15	48.98	61.81	0.76	0.51
Fold 3	57.23	57.97	52.68	62.81	0.79	0.54
Fold 4	49.59	58.36	41.22	53.20	0.74	0.42
Fold 5	57.23	56.80	51.09	63.02	0.78	0.53
Average	55.76	58.91	49.48	61.60	0.77	0.51

**Table 4 cancers-15-02974-t004:** Top-1 accuracy of network with gender and age classifier.

Method of Gender and Age Classifier	Top-1 Accuracy (%)
Naive Bayes	37.96 (35.87~40.05)
Logistic Regression	32.05 (20.16~33.94)
Support Vector Classifier	33.90 (31.92~35.88)
K-Nearest Neighbors	23.06 (21.35~25.77)
Random Forest	22.76 (21.98~23.62)

## Data Availability

The data can be shared up on request.

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
