# Peer review of "Prediction of Primary Tumor Sites in Spinal Metastases Using a ResNet-50 Convolutional Neural Network Based on MRI"

_cancers, 2023, doi:10.3390/cancers15112974_

Round 1

Reviewer 1 Report

The manuscript “Prediction of Primary Tumor Sites in Spinal Metastases Using a ResNet-50 Convolutional Neural Network Based on MRI” by  Ke Liu et al. aimed to investigate the feasibility and evaluate the performance of a Res- Net-50 CNN model based on MRI in predicting primary tumor sites in spinal metastases.

Below are my comments and remarks regarding the manuscript:

1. The introduction is correctly described

2. Why was MRI with contrast not used?

3. The results could be described in more detail

4. Limitation have been well described

Author Response

Dear Reviewer,

We would like to express our gratitude for your review and insightful comments, which have greatly assisted us in enhancing the quality of our manuscript. We have provided a detailed response to each point raised, and all substantial modifications have been highlighted in the annotated version of the manuscript. We believe that your suggestions have significantly improved the quality of the revised manuscript, and we sincerely hope that you find the revisions satisfactory for publication. If there are any additional questions or points requiring clarification, please do not hesitate to let us know. We assure you of our prompt response.

Comment 1: The introduction is correctly described

Reply 1: We appreciate your positive comment on the introduction of our paper. We strive to present a clear and concise introduction that sets the context for the rest of the study.

Comment 2: Why was MRI with contrast not used?

Reply 2: The usage of MRI without contrast in this study was primarily due to the nature of the available data. Using non-contrast MRI scans was a decision made based on the datasets we had access to. We acknowledge that contrast-enhanced MRI scans could potentially provide more detailed imaging data, and we also discussed this issue in the third point of the limitations.

Comment 3: The results could be described in more detail

Reply 3: We agree with your comment on the description of the results. In our revision, we add the range based on 95% confidence interval. It is a measure of the precision or uncertainty of the estimate. We have added the results of the 5-fold cross-validation in Table 3. The evaluation results demonstrate stability across all folds. In order to address these points, we have supplemented the second paragraph of the “2.6 Training and Evaluation” section and the third paragraph of the “3.2 Performance of the ResNet-50 Model” section with additional explanations.

Comment 4: Limitation have been well described

Reply 4: We're glad to hear that the limitations of our study were well described. We believe that acknowledging the limitations is crucial for the transparency of the research and helps guide future work in this area.

Thank you again for your valuable feedback. We look forward to further improving our manuscript based on your comments.

Best regards,

Ke Liu

Reviewer 2 Report

I think that this is an important topic in spinal metastases care to deal with patients with cancer of unknown primary. If this approach has the potential to identify top candidates for the unknown primary, it may lead to the diagnostic process with less time-consuming and less expensive. Authors developed the ResNet-50 deep learning model based on conventional neural network and MRI imaging technique, which can help the categorization of the metastatic tumor type for patients with unknown primary sites on spinal metastases. As a reviewer, I would like to have several comments on methodology and suggest several issues for the readers.

1. A total 295 patients (from 142 from lung cancer to 28 from prostate cancer) were evaluated to apply a deep learning model using ResNet-50 CNN to classify unknown primary tumor sites. 28 patients from prostate cancer look very small. I think authors should explain the righteousness of their samples and include at least post hoc sample-size calculations for prove the better predictive power of your study.  

2. Authors applied training (90%) and test (10%) split data. If authors apply cross-validation based approach (repeat 10 different combinations of training and test splits) and compare/aggregate the results, it may provide better prediction results. It might be interesting to see the variabilities among the different splits.

3. Authors said that they removed the gender and age classifiers in the final results because the results of classifiers damage the overall result when combined. Can you explain what damage came from the incorporation of the classifiers? As authors mentioned, age and gender are important baseline demographic factors for diagnosis. From Table 1, thyroid cancer shows highly imbalance on gender and age and kidney cancer shows the imbalance of gender distribution. Authors might look at the results with the classifiers.

4. Please explain the parameters in equation (1), nc, p, p^, WBCE(p,p^), and provide more information.. where did this formula come from, what does WBCE(p,p^) mean? And how this be used for balancing? etc.

5. For Table 2, the readers might be interested in the interpretation of the numbers. I suggest to provide more information about the numbers.. like the ranges… the higher is the better?... how can we differentiate the numbers between 5-class and 3-class -  is it possible to test the comparison?

6. In Table 2, sensitivities are quite low. Is this fine for the potential of a diagnostic tool? And even sensitivities are low, the corresponding AUCs are relatively high. The readers may need to know why this happens.

7. For Table 3, random forests (RF) methods have usually shown better performance in other study areas. But RF shows the lowest here. Any specific reason?  

Author Response

Dear Reviewer,

We would like to express our gratitude for your review and insightful comments, which have greatly assisted us in enhancing the quality of our manuscript. We have provided a detailed response to each point raised, and all substantial modifications have been highlighted in the annotated version of the manuscript. We believe that your suggestions have significantly improved the quality of the revised manuscript, and we sincerely hope that you find the revisions satisfactory for publication. If there are any additional questions or points requiring clarification, please do not hesitate to let us know. We assure you of our prompt response.

Comment 1: A total 295 patients (from 142 from lung cancer to 28 from prostate cancer) were evaluated to apply a deep learning model using ResNet-50 CNN to classify unknown primary tumor sites. 28 patients from prostate cancer look very small. I think authors should explain the righteousness of their samples and include at least post hoc sample-size calculations for prove the better predictive power of your study.

Reply 1: As you correctly pointed out, the dataset is unbalanced among the categories. Our model handles this problem by introducing the weighted loss function. A detailed description of this method can be found in the second paragraph of the “2.3 Data Pre-processing” Section. The 3-class classification results in Table 2 support that our model could still perform well even with more severe unbalance caused by the decreased number of categories. In addition, we have further revised the first limitation of this paper to provide a more comprehensive explanation of this issue.

Comment 2: Authors applied training (90%) and test (10%) split data. If authors apply cross-validation based approach (repeat 10 different combinations of training and test splits) and compare/aggregate the results, it may provide better prediction results. It might be interesting to see the variabilities among the different splits.

Reply 2: We have taken your advice and included the results of the 5-fold cross-validation in Table 3. The evaluation results demonstrate stability across all folds. In order to address these points, we have supplemented the second paragraph of the “2.6 Training and Evaluation” section and the third paragraph of the “3.2 Performance of the ResNet-50 Model” section with additional explanations. While it is true that a higher value of k could be used for cross-validation, repeating the experiments becomes time-consuming. However, if you have any other concerns, we are open to considering higher values of k for future experiments.

Comment 3: Authors said that they removed the gender and age classifiers in the final results because the results of classifiers damage the overall result when combined. Can you explain what damage came from the incorporation of the classifiers? As authors mentioned, age and gender are important baseline demographic factors for diagnosis. From Table 1, thyroid cancer shows highly imbalance on gender and age and kidney cancer shows the imbalance of gender distribution. Authors might look at the results with the classifiers.

Reply 3: Gender and age are important hand-craft features for diagnosis based on previous research. Here, we are not trying to deny this assumption but want to show that our model can perform very well without designed features. Convolution neural networks make predictions based on high-dimensional information that may contain humans’ known and unknown features. The gender and age could be redundant here because the model may have already got sufficient information to make the correct prediction. Explainable AI is out of the scope of this research. But we re-paragraph the related content in the fourth paragraph of the “3.2 Performance of the ResNet-50 Model” section to make it clearer and avoid misunderstanding.

Comment 4: Please explain the parameters in equation (1), nc, p, p^, WBCE(p,p^), and provide more information. where did this formula come from, what does WBCE(p,p^) mean? And how this be used for balancing? etc.

Reply 4: We have updated the explanation about equation (1). The weighted cross entropy aims to give the samples in the minority category higher weights. It suppresses unbalance by forcing rare samples to contribute more to the loss function.

Comment 5: For Table 2, the readers might be interested in the interpretation of the numbers. I suggest to provide more information about the numbers. like the ranges… the higher is the better?... how can we differentiate the numbers between 5-class and 3-class -  is it possible to test the comparison?

Reply 5: Thank you very much for your feedback. It has brought to our attention that our explanation of the meaning of the 3-class and 4-class results in Table 2 was not clear enough, leading to potential confusion for readers. We would like to clarify that the 3-class results are the average result among all possible combinations of 3 classes out of 5. We use our model to do  a series of 3-class classification by only replacing the toppest layer. We provided a clearer explanation in the first paragraph of the “3.2 Performance of the ResNet-50 Model”. We also add the range based on 95% confidence interval.

Comment 6: In Table 2, sensitivities are quite low. Is this fine for the potential of a diagnostic tool? And even sensitivities are low, the corresponding AUCs are relatively high. The readers may need to know why this happens.

Reply 6: Sensitivity is static while AUC is dynamic. They are relevant but do not necessarily change at the same pace. Our model is a primary proposal to solve this problem, which clearly has the potential to improve the precision as a diagnostic tool. We will focus on making it more accurate in our following research. We conducted additional discussions in the limitations section of the article.

Comment 7: For Table 3, random forests (RF) methods have usually shown better performance in other study areas. But RF shows the lowest here. Any specific reason?

Reply 7: Random forest is a powerful model that is capable to handle many tasks, as well as other classic machine learning algorithms we considered. But these models also fail in many tasks. Why random forests cannot give promising performance under this problem-setting is not our interest. We are happy if other researches could analyze this deeper in the future.

Thank you again for your valuable feedback. We look forward to further improving our manuscript based on your comments.

Best regards,

Ke Liu

Round 2

Reviewer 1 Report

I have no more comments

Author Response

Dear Reviewer,

Thank you for taking the time to review our manuscript and for your positive feedback. We're glad to hear that you have no further comments on our work. We appreciate the insightful comments and suggestions you provided, which have significantly contributed to improving the quality of our paper.

We look forward to your final decision and are fully committed to making any further modifications if necessary, to meet the journal's publication standards. Please do not hesitate to contact us if there are any other issues or questions regarding our study.

Thank you once again for your careful consideration of our work.

Sincerely,

Ke Liu

Reviewer 2 Report

I think that authors incorporated most of my comments and suggestions. But I think a couple of more comments related to methodology with this revised version should be confirmed from the authors.

1. I think that authors should explain the compatibility of the sample sizes used in machine learning methods. Most diseases have small samples (28 for prostate, 34 for thyroid, 41 for mammary..). Authors need to explain how the deep learning method with small samples can produce reliable results. 

2. In this study, the sensitivity can be interpreted as the probability that the cancer(s) actually exists in spinal  metastases when the cancer(s) are identified by the proposed ResNet-50 method. In Table 2, the overall sensitivity and AUC levels are not high. I think that authors need to explain whether these numbers can be accepted as a reliable tool with sufficient power to be able to detect the unknown cancer in the metastases. Also please add a column of specificity in Table 2. 

Author Response

Dear Reviewer,

We wish to express our deep appreciation for your comprehensive review and valuable feedback. Your comments and suggestions have once again guided us in making crucial improvements to our manuscript. We have carefully addressed each point you raised and have made corresponding modifications, which are highlighted in the revised version of the manuscript. We firmly believe that under your expert guidance, our manuscript has undergone considerable enhancement, bringing it closer to the standard required for publication. We genuinely hope that our responses and revisions meet your approval. Should you have any further questions or require additional clarification on any point, please do not hesitate to bring it to our attention. We remain at your disposal for any further communication, promising a swift response to any inquiries.

Comment 1: I think that authors should explain the compatibility of the sample sizes used in machine learning methods. Most diseases have small samples (28 for prostate, 34 for thyroid, 41 for mammary..). Authors need to explain how the deep learning method with small samples can produce reliable results.

Reply 1: For each patient, there are at least 13 MRI images used for training and evaluation. Our model takes a single image instead of a single patient as a sample. Therefore, the real capacity of our dataset is higher than the number of patients multiplied by 13. Take the most minor category prostate cancer as an example, there are more than 28*13=364 images, which are sufficient to train and validate a deep-learning model. Nevertheless, the deep learning model will be more credible with a larger dataset. We will consider collecting more valid samples and making a larger dataset in the future.  

Comment 2: In this study, the sensitivity can be interpreted as the probability that the cancer(s) actually exists in spinal  metastases when the cancer(s) are identified by the proposed ResNet-50 method. In Table 2, the overall sensitivity and AUC levels are not high. I think that authors need to explain whether these numbers can be accepted as a reliable tool with sufficient power to be able to detect the unknown cancer in the metastases. Also please add a column of specificity in Table 2.

Reply 2: The overall sensitivity and AUC levels of our model are not higher compared with previous medical image recognition models. This is because we have a more challenging task which cannot be solved by even an experienced doctor. The problem we are facing here is different from common classification tasks, such as whether there is a tumor in the image, that is finely solved by humans. From this point, we believe our model is meaningful for exploring a possible way to solve this problem. Regarding your concern about the reliability of our model as a tool, we think it could work as an assistance tool because it has given a relatively reasonable prediction compared with the one made by humans. The doctor could then arrange more tests first at the highly suspended tumor site instead of making a random selection. And of course, as we mentioned before, optimizing this tool is on our ongoing plan. Thank you for raising your concern about this point, we have added more relevant content in the discussion to share our thoughts and future plans with the readers. Your suggestion of adding a column for specificity in Table 2 is well received, and we have incorporated it in the revised version of our manuscript.

Thank you again for your valuable feedback. We look forward to further improving our manuscript based on your comments.

Best regards,

Ke Liu

Round 3

Reviewer 2 Report

I don't have any more issues on this version. Thanks for your fully follow-up on the comments.